# Improving the coverage of credible sets in Bayesian genetic fine-mapping

**Anna Hutchinson**[1]*, **Hope Watson**[1], **Chris Wallace**[1,2]*

**1** MRC Biostatistics Unit, Cambridge Institute of Public Health, Cambridge, United Kingdom, **2** Cambridge Institute for Therapeutic Immunology and Infectious Disease, University of Cambridge, Cambridge, United Kingdom

* anna.hutchinson@mrc-bsu.cam.ac.uk (AH); cew54@cam.ac.uk (CW)

**Data Availability Statement:** All relevant data are within the manuscript and its Supporting Information files.

**Funding:** AH is supported by The Engineering and Physical Sciences Research Council (EPSRC) [EP/

## Abstract

Genome Wide Association Studies (GWAS) have successfully identified thousands of loci associated with human diseases. Bayesian genetic fine-mapping studies aim to identify the specific causal variants within GWAS loci responsible for each association, reporting credible sets of plausible causal variants, which are interpreted as containing the causal variant with some "coverage probability". Here, we use simulations to demonstrate that the coverage probabilities are over-conservative in most fine-mapping situations. We show that this is because fine-mapping data sets are not randomly selected from amongst all causal variants, but from amongst causal variants with larger effect sizes. We present a method to re-estimate the coverage of credible sets using rapid simulations based on the observed, or estimated, SNP correlation structure, we call this the "adjusted coverage estimate". This is extended to find "adjusted credible sets", which are the smallest set of variants such that their adjusted coverage estimate meets the target coverage. We use our method to improve the resolution of a fine-mapping study of type 1 diabetes. We found that in 27 out of 39 associated genomic regions our method could reduce the number of potentially causal variants to consider for follow-up, and found that none of the 95% or 99% credible sets required the inclusion of more variants—a pattern matched in simulations of well powered GWAS. Crucially, our method requires only GWAS summary statistics and remains accurate when SNP correlations are estimated from a large reference panel. Using our method to improve the resolution of fine-mapping studies will enable more efficient expenditure of resources in the follow-up process of annotating the variants in the credible set to determine the implicated genes and pathways in human diseases.

## Author summary

Pinpointing specific genetic variants within the genome that are causal for human diseases is difficult due to complex correlation patterns existing between variants. Consequently, researchers typically prioritise a set of plausible causal variants for functional validation—these sets of putative causal variants are called "credible sets". We find that the probabilistic interpretation that these credible sets do indeed contain the true causal variant is

R511870/1] and GlaxoSmithKline. CW is supported by the Wellcome Trust [WT107881] and the Medical Research Council (MRC) [MC_UU_00002/4]. The funders had no role in study design, data collection and analysis, decision to publish, or preparation of the manuscript.

**Competing interests:** The authors have declared that no competing interests exist.

variable, in that the reported probabilities often underestimate the true coverage of the causal variant in the credible set. We have developed a method to provide researchers with an "adjusted coverage estimate" that the true causal variant appears in the credible set, and this has been extended to find "adjusted credible sets", allowing for more efficient allocation of resources in the expensive follow-up laboratory experiments. We used our method to reduce the number of genetic variants to consider as causal candidates for follow-up in 27 genomic regions that are associated with type 1 diabetes.

## Introduction

Genome-Wide Association Studies (GWAS) have identified thousands of disease-associated regions in the human genome, but the resolution of these regions is limited due to linkage disequilibrium (LD) between variants [1]. Consequently, GWAS identifies multiple statistical, but often non-causal, associations at common genetic variants (typically SNPs) that are in LD with the true causal variants. Follow-up studies are therefore required for the prioritisation of the causal variants within these regions, which is an inherently difficult problem due to convoluted LD patterns between hundreds or thousands of SNPs. Consequently, fine-mapping studies prioritise a set of variants most likely to be causal in each risk locus. Laboratory functional studies or large-scale replication studies may then be used to identify the true causal variants within these sets, which can then be linked to their target genes to better understand the genetic basis of human diseases [2, 3].

Early statistical approaches for fine-mapping tended to focus on the SNP in the region with the smallest *P* value, called the lead-SNP. However, it is generally acknowledged that this SNP may not be the causal variant in a given region due to correlations with the true causal variants [1, 4]. Studies may therefore prioritise the lead-SNP before extending the analysis to include either variants in high LD with this SNP or the top *k* variants with the highest evidence of association [5].

Fine-mapping is analogous to a variable selection problem with many highly correlated variables (the SNPs) [6]. As such, methods such as penalised regression have also been adopted for fine-mapping, with the aim of choosing the variables representing the variants most likely to be causal for inclusion in the final model [7, 8]. Yet these methods ultimately select one final model and lack probabilistic quantification for this selected model.

Bayesian approaches for fine-mapping [9–13] use posterior probabilities of causality (PPs) to quantify the evidence that a variant is causal for a given disease, and these can be meaningfully compared both within and across studies. The standard Bayesian approach for fine-mapping was developed by Maller et al. (2012) and assumes a single causal variant per genetic region to prioritise an "($\alpha \times 100$)% credible set" of putative causal variants. This is derived by ranking variants based on their PPs and summing these until a threshold, $\alpha$, is exceeded—with the variants required to exceed this threshold comprising the credible set.

The "coverage" of credible sets refers to the probability that the causal variant is contained in the credible set. Coverage is typically understood to refer to that probability taken over all possible realisations of the data. However, fine-mapping is conventionally only performed on a subset of genomic regions that contain at least one SNP with a *P* value of association below some threshold (for example genome-wide significance). We therefore examine *conditional coverage* here, defined as the probability that the causal variant is contained in the credible set, conditional on some aspect of the observed data, for example that it has been selected for fine-mapping.

In the literature, credible sets are interpreted as having good frequentist conditional coverage of the causal variant [9, 14, 15], although there is no mathematical basis for this [6]. For example, researchers often state that an $(\alpha \times 100)$% credible set contains the causal variant with $(\alpha \times 100)$% probability [16–20] or with probability $\geq (\alpha \times 100)$% [6, 21, 22]. More specifically, they may be interpreted as containing the causal variant with probability equal to the sum of the PPs of the variants in the credible set [23], for which the threshold forms a lower bound. A simulation study found that the conditional coverage of the causal variant in 95% and 99% credible sets varied with the power to detect the signal (Fig S1 in [1]), implying that inferring the frequentist conditional coverage estimate of these Bayesian credible sets may not be as straightforward as the literature suggests.

In this work, we investigate the accuracy of standard conditional coverage estimates reported in the Bayesian single causal variant fine-mapping literature. We develop a new method to re-estimate the conditional coverage of these credible sets, deriving an "adjusted coverage estimate" and extending this to construct an "adjusted credible set". We validate our method through simulations and demonstrate its improved performance relative to standard conditional coverage estimates reported in the literature.

Our method is available as a CRAN R package, `corrcoverage` (https://github.com/annahutch/corrcoverage; https://cran.r-project.org/web/packages/corrcoverage/index.html), which was used to decrease the size of the standard 95% credible sets for 27 out of 39 genomic regions that are associated with type 1 diabetes. Crucially, our method requires only summary-level data and remains accurate when SNP correlations are estimated from a reference panel (such as the UK10K project [24]).

## Results

### Accuracy of claimed coverage estimates in single causal variant fine-mapping literature is variable

To investigate the true conditional coverage of the causal variant in Bayesian credible sets, we simulated a variety of single causal variant association studies using 1006 European haplotypes or 1322 African haplotypes from the 1000 Genomes Phase 3 data set [25].

Briefly, SNPs were sampled from various genomic regions with differing LD structures. In each simulation, causality was randomly allocated to one of the variants in the region with an additive phenotypic effect. Sample sizes were also varied across simulations (NN: number of cases = number of controls = 5000, 10000 or 50000). We calculated the frequentist empirical estimate of the true conditional coverage for each simulated credible set as the proportion of 5000 replicate credible sets that contained the simulated causal variant.

Bayesian fine-mapping frameworks require specification of a prior distribution on the effect size of the causal variant, conventionally $N(0, W)$ with $W = 0.2^2$ for a log odds ratio in a case-control study. We therefore simulated association studies that reflect the underlying Bayesian single causal variant fine-mapping model exactly, such that in each simulation, causality was randomly selected across all SNPs and the causal effect size ($\beta = $ log odds ratio) was sampled from $N(0, 0.2^2)$ and then fixed at its sampled value to calculate an estimate of the empirical conditional coverage. However, we argue that in any individual dataset the effect size at the causal variant is sampled from a point distribution with unknown but fixed mean, and that it is this mean which may be considered to be sampled from $N(0, 0.2^2)$.

We examined the distribution of genome-wide significant lead-SNP effect sizes from data deposited in the GWAS catalogue [26] to select representative point distributions to use in our analysis (S1 Fig). Overall, in each of our simulations effect sizes were either (i) sampled from the prior distribution, $\beta \sim N(0, 0.2^2)$ (ii) fixed at $\beta = log(1.05)$ or (iii) fixed at $\beta = log(1.2)$.

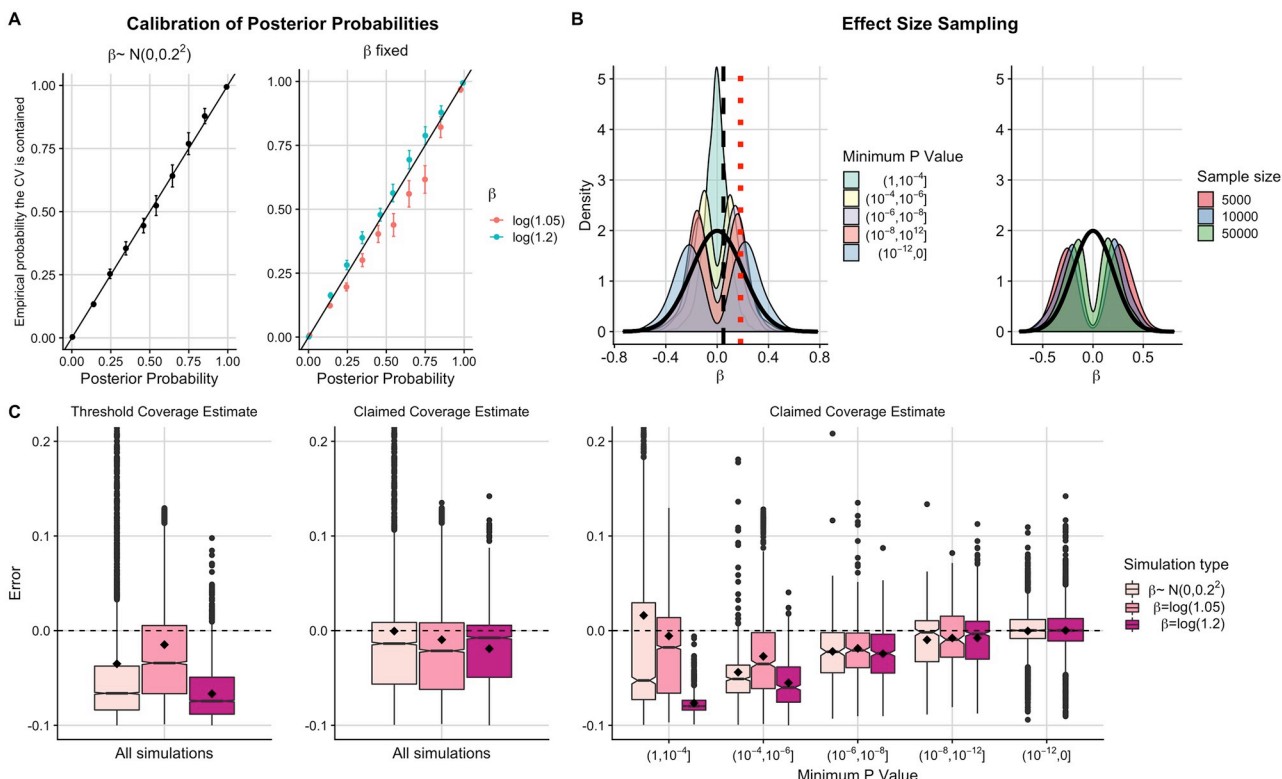

**Fig 1. Investigating the error of conditional coverage estimates in Bayesian single causal variant fine-mapping.** (A) Calibration of posterior probabilities for simulations where (left) $\beta \sim N(0, 0.2^2)$ (right) $\beta = log(1.05)$ or $\beta = log(1.2)$. Posterior probabilities from 13000 simulations were binned into 10 equally sized bins. The 'empirical probability the CV is contained' is the proportion of causals in each posterior probability bin, with $+/-1.96 \times$ standard error shown. (B) Distribution of sampled $\beta$ values in simulations where $\beta \sim N(0, 0.2^2)$ for (left) each P value bin or (right) each sample size bin in simulations where $P_{min} < 10^{-8}$. Thick black curve is density function for $N(0, 0.2^2)$. Black dashed line shows $\beta = log(1.05)$ and red dotted line shows $\beta = log(1.2)$. (C) Box plots (median and IQR with mean marked by black diamond) of error in (left) threshold and (middle) claimed coverage estimates when averaged across all 5000 simulations or (right) when simulations have been binned by the minimum P value in the region. Error is defined as estimated conditional coverage—empirical conditional coverage and empirical conditional coverage is the proportion of 5000 replicate credible sets that contain the causal variant. The two simulations where $\beta = log(1.05)$ that were in the $(10^{-12}, 0]$ bin were manually removed as a box plot could not be formed.

A common feature of Bayesian statistical inference is that if one simulates from the prior used in the underlying model, then the resultant posterior inferences will be accurate. This is reflected in our results, where the PPs are well calibrated in simulations where the effect size at the causal variant is sampled from the prior normal distribution (Fig 1A; left).

However, we found that some bias is introduced when the effect sizes are sampled from a point distribution, as in a standard fine-mapping study. Specifically, the PPs tend to be anti-conservatively biased in low effect sizes and conservatively biased in higher effect sizes (Fig 1A; right).

We then used these PPs to generate 90% credible sets of putative causal variants, following the standard Bayesian single causal variant fine-mapping procedure (see Methods and [9]). We investigated the accuracy of setting the conditional coverage estimate of the credible set equal to both the threshold ("threshold coverage estimate") [16–20] and the sum of the PPs of the variants in the set ("claimed coverage estimate") [23]. We found that there is accumulative bias carried through from SNP PPs to conditional credible set coverage estimates.

When averaging results over all simulations, threshold coverage estimates tended to be conservatively biased for credible sets, even when effect sizes are sampled from the prior

distribution (Fig 1C; left). However, low effect sizes can also lead to anti-conservative estimates in regions with differing LD structures (S2 and S3 Figs) and in low powered studies (S4 Fig). The claimed coverage estimates are unbiased when effect sizes are sampled from the prior distribution, but with large variability in the direction of conservative estimates. On the contrary, when effect sizes are sampled from point distributions, the claimed coverage estimates are systematically biased (Fig 1C; middle), a result that is consistent across various LD structures and credible set thresholds (S2 and S3 Figs).

Researchers typically select regions to fine-map depending on the strength of evidence of an association in that region, which is usually quantified by a $P$ value (for example, regions reaching genome-wide significance). We therefore further scrutinise the accuracy of the conditional coverage estimates by binning simulations by minimum P value ($P_{min}$) in the region. Each P value bin encompasses simulations which sample different parts of the prior effect size distribution, such that the distribution of effect sizes for each P value bin no longer resemble the conventional $N(0, 0.2^2)$ prior (Fig 1B).

We found that claimed coverage estimates are systematically biased in representatively powered scenarios where fine-mapping is usually performed ($P_{min} > 10^{-12}$) regardless of the effect size sampling method (Fig 1C; right). This bias is mostly conservative, but may be anti-conservative in low powered studies (demonstrated by splitting simulations up by sample size, S4 Fig). Notably, even when the effect sizes are sampled from the prior, the claimed coverage estimates are anti-conservatively biased in low powered simulations ($P_{min} > 10^{-4}$), conservatively biased in intermediately powered simulations ($10^{-12} < P_{min} < 10^{-4}$) and unbiased in very high powered simulations ($P_{min} < 10^{-12}$).

These findings are consistent across various LD patterns and credible set thresholds, with greatest variability detected between estimates in high LD regions (S2 and S3 Figs). To investigate the impact of LD further, we repeated the analysis, varying LD patterns over a much larger population (7562 European UK10K haplotypes) and averaging the results over a range of LD patterns (see Methods). We found that the results were similar to those shown in Fig 1 (S5 Fig).

In conclusion, the probabilities that the causal variant is contained within the credible set in intermediately powered single causal variant fine-mapping studies are typically too low, and researchers can afford to be "more confident" that they have captured the true causal variant in their credible set. We demonstrate that this bias is due to researchers selecting a biased sample of regions to fine-map and that this bias is increased when considering causal variant effect sizes as being sampled from various point distributions.

## Adjusted coverage estimate improves empirical calibration of credible sets

We developed a new estimator for the true conditional coverage of the causal variant in credible sets, called the "adjusted coverage estimate", which is based on learning the bias in the system by repeatedly simulating summary GWAS data from the same MAF and LD structure as the observed data. We derive an estimate for the expected $Z$ score at the (unknown) causal variant as a weighted average of absolute $Z$ scores taken across SNPs in the region, using the observed PPs as weights (S6 Fig) (see Methods for detailed derivation).

For each of the simulated credible sets, we found that the adjusted coverage estimates were better empirically calibrated than the claimed coverage estimates in simulations that are representative of those considered for fine-mapping ($P_{min} < 10^{-6}$) (Fig 2; S2, S3 and S5 Figs). Particularly, the median and mean error of the adjusted coverage estimates decreases for $P_{min} < 10^{-6}$, and the variability between estimates also decreases even in simulations where the claimed coverage estimates are unbiased ($P_{min} < 10^{-12}$). Our findings are robust to the choice of value for $W$ in the causal variant effect size prior, $N(0, W)$ (S7 Fig).

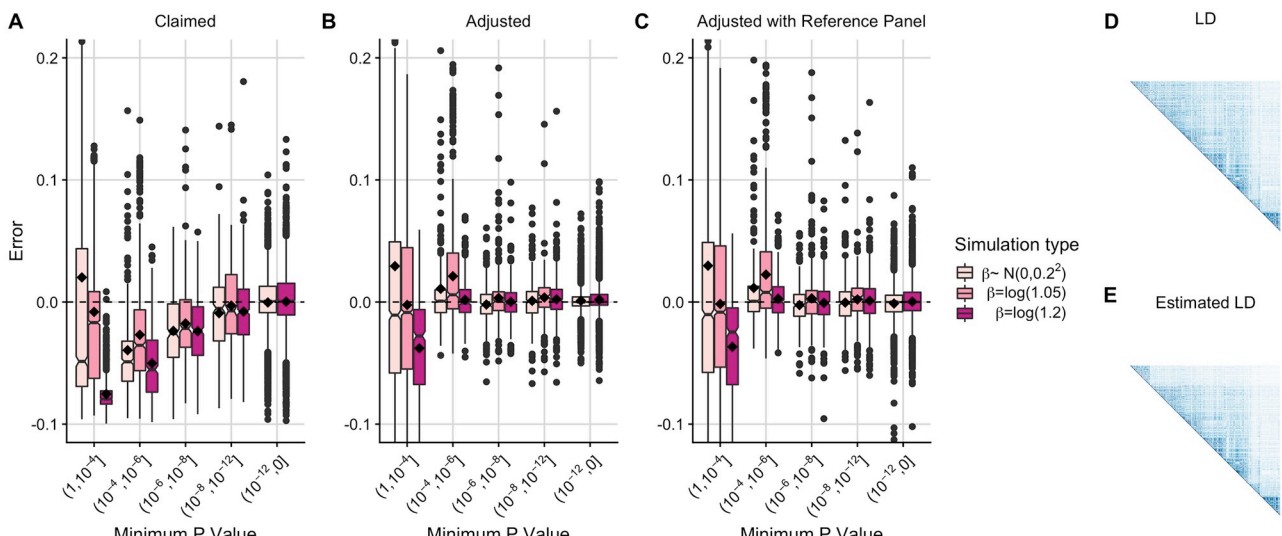

**Fig 2. Error of conditional coverage estimates for 90% credible sets including using a reference panel to approximate MAFs and SNP correlations.** Box plots (median and IQR with mean marked by black diamond) of error in conditional coverage estimates from 5000 simulated 90% credible sets. Error is calculated as estimated conditional coverage–empirical conditional coverage and empirical conditional coverage is the proportion of 5000 additional simulated 90% credible sets that contain the causal variant. The two simulations with $\beta = log(1.05)$ that were in the $(10^{-12}, 0]$ bin were manually removed as a box plot could not be formed. (A) Claimed coverage estimate (the sum of the posterior probabilities of causality for the variants in the credible set) (B) Adjusted coverage estimate using MAFs and SNP correlations from the original (1000 Genomes) data (C) Adjusted coverage estimate using UK10K data to approximate MAFs and SNP correlations (D) Graphical display of SNP correlations in 1000 Genomes data (E) Graphical display of the estimated SNP correlations using UK10K data.

**Adjusted coverage robust to MAF and LD estimated from a reference panel.** Our method relies on MAF and SNP correlation data to simulate GWAS summary statistics representative of the observed GWAS data. So far we have assumed that this information is available from the GWAS samples, but due to privacy concerns this is not generally the case. We therefore evaluated the performance of our adjustment method when using independent reference data to estimate MAFs and SNP correlations. We applied our method to credible sets simulated from the European 1000 Genomes data using either MAFs and SNP correlations from the original (1000 Genomes) data (Fig 2B) or MAFs and SNP correlations estimated from a reference panel (UK10K) (Fig 2C). We found that the adjusted coverage estimates remained accurate in either case (Fig 2C, S8 Fig).

**Adjusted coverage robust to departures from single causal variant assumption.** The Bayesian approach for fine-mapping described by Maller et al. assumes a single causal variant per genomic region, which may be unrealistic [27]. Using simulated data with 2 causal variants, and defining conditional coverage as the frequency with which a credible set contained at least 1 causal variant, we found that the adjusted coverage estimates tended to have smaller error than the claimed coverage estimates for causal variants in low LD ($r^2 < 0.01$, Fig 3A). When the 2 causal variants are in high LD ($r^2 > 0.7$), the adjusted coverage estimates are still generally more accurate than the claimed coverage estimates, although both tend to underestimate the true conditional coverage (and are thus conservative) (Fig 3B). Whilst we do not recommend using the Bayesian single causal variant fine-mapping method (and therefore our adjustment) when the single causal variant assumption is violated, we show that if this is the case then our method performs no worse than the standard method, when considering capturing at least 1 causal variant as most relevant.

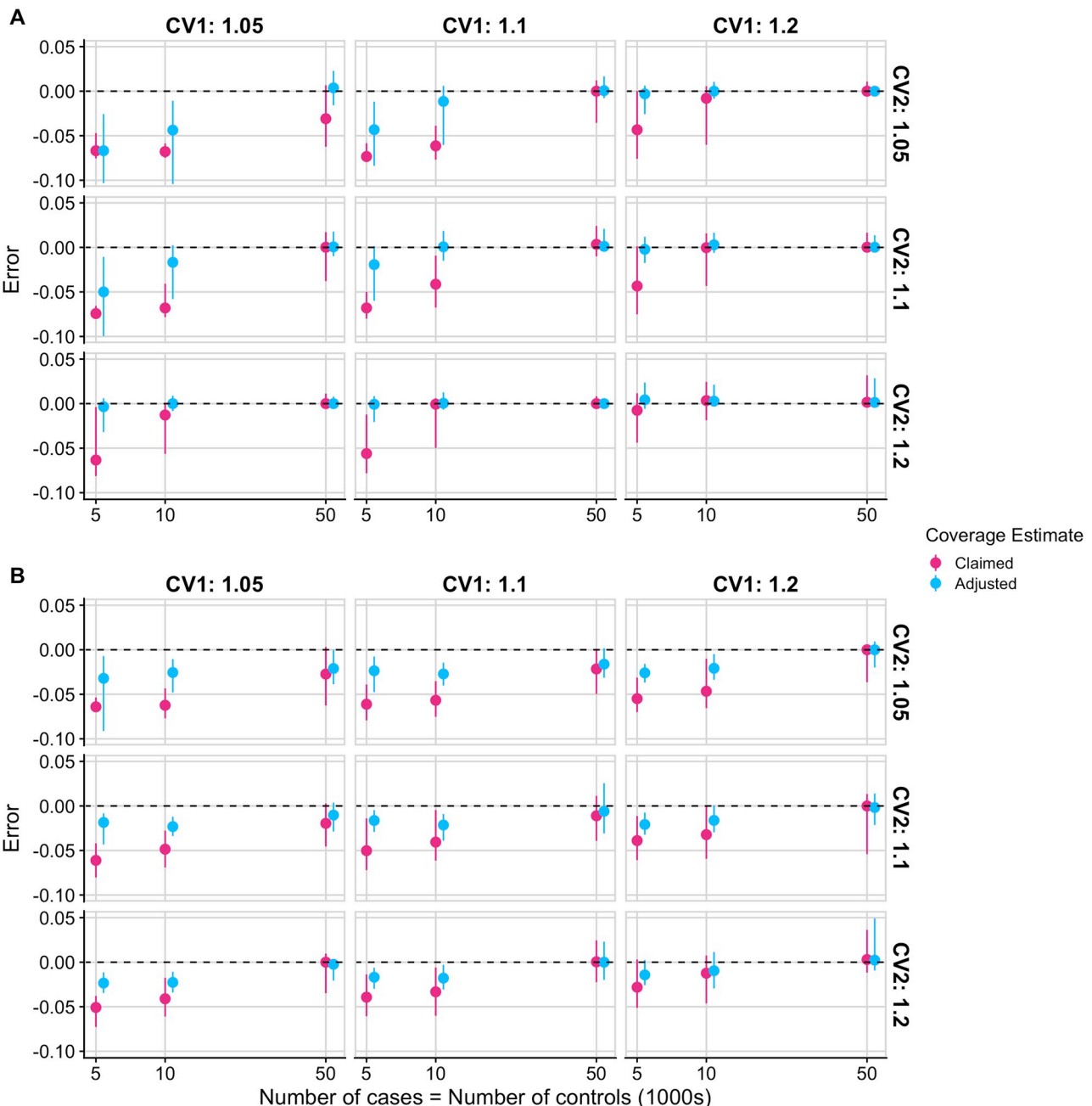

**Fig 3. Error of conditional coverage estimates for 90% credible sets in regions with 2 causal variants.** Error is calculated as estimated conditional coverage–empirical conditional coverage where empirical conditional coverage is the proportion of 5000 additional simulated 90% credible sets that contain at least one of the 2 causal variants and estimated conditional coverage is the claimed or adjusted coverage estimate as defined in the text. The median error and interquartile range of claimed and adjusted coverage estimates of 90% credible sets from 5000 simulated regions with 2 causal variants that are (A) in low LD ($r^2 < 0.01$) (B) in high LD ($r^2 > 0.7$). Faceted by odds ratio values at the causal variants.

### Adjusted credible sets

Obtaining an accurate conditional coverage estimate that the causal variant appears in the credible set is useful in its own right, but it is also beneficial to obtain an "adjusted credible set"—that is, the smallest set of variants required such that the adjusted coverage estimate of

the resultant credible set achieves some desired conditional coverage of the causal variant. For example, discovering that a 90% credible set actually has 99% conditional coverage of the causal variant is useful, but an obvious follow-up question is "What variants do I need such that the conditional coverage is actually 90%?".

We explored this using an example simulated GWAS across 200 SNPs with the effect size fixed at $\beta = log(1.2)$. The 90% credible set, constructed using the standard Bayesian approach, contained 8 variants and had a claimed coverage estimate of 0.903. The adjusted coverage estimate of this credible set was 0.969 and the estimated empirical coverage was 0.972.

We used the root bisection method [28] to iteratively search for the smallest threshold value that yields a credible set with accurate conditional coverage of the causal variant. In this example, we found that an adjusted 90% credible set could be constructed using a threshold value of 0.781. This adjusted credible set had a adjusted coverage estimate of 0.905 (empirical estimated conditional coverage of 0.907) and reduced in size from 8 to 4 variants, with the 4 variants removed from the credible set holding a small proportion of the total posterior probability (Fig 4).

Simulations confirmed that the empirical conditional coverage probabilities of adjusted credible sets created using this method are accurate, such that on average the empirical estimate of the true conditional coverage of an adjusted 90% (or 95%) credible set is indeed 90% (or 95%) (S9 Fig).

**corrcoverage R package.** We created a CRAN R package, `corrcoverage` (https://annahutch.github.io/corrcoverage/; https://cran.r-project.org/web/packages/corrcoverage/index.html), that uses marginal summary statistics to derive adjusted coverage estimates and adjusted credible sets. The functions to calculate adjusted coverage estimates are computationally efficient, taking approximately 1 minute for a 1000 SNP region (using one core of an Intel Xeon E5-2670 processor running at 2.6GHz; S10 Fig).

The functions to derive adjusted credible sets require only the summary statistics needed to derive the adjusted coverage estimate ($Z$ scores, MAFs, sample sizes and SNP correlation matrix) plus some user-specified desired coverage. Users are able to customise the optional arguments to suit both their accuracy requirements and computational constraints. The algorithm then works iteratively such that the threshold and the adjusted coverage estimate of each tested credible set is displayed, until the smallest set of variants with the desired conditional coverage is established, offering researchers an easy tool to improve the resolution of their credible sets.

## Impact of adjusting credible sets in a GWAS

We applied our adjustment method to association data from a large type 1 diabetes (T1D) genetic association study consisting of 6,670 cases and 12,262 controls [29]. In the original study, 99% credible sets are found for 40 genomic regions. Here we focus on 95% credible sets as these best illustrate the utility of our method due to the greater margin for error, and we exclude the *INS* region with lead SNP rs689 which failed QC in the Immunochip (and for which additional genotyping data was used in the original study).

The results match our previous findings—that the claimed coverage estimates are often lower than the adjusted coverage estimates (Fig 5). We found that the size of the 95% credible set could be reduced in 27 out of the 39 regions, without the use of any additional data (S1 and S2 Tables). Similarly, we found that the size of the 99% credible set could be reduced in 27 out of the 39 regions (S11 Fig, S2 File, S3 and S4 Tables).

Generally, credible sets were more likely to be subject to change where minimum $P$ values were larger (S12A Fig). In 16 "high-powered" regions (minimum $P < 10^{-12}$), 6 were not

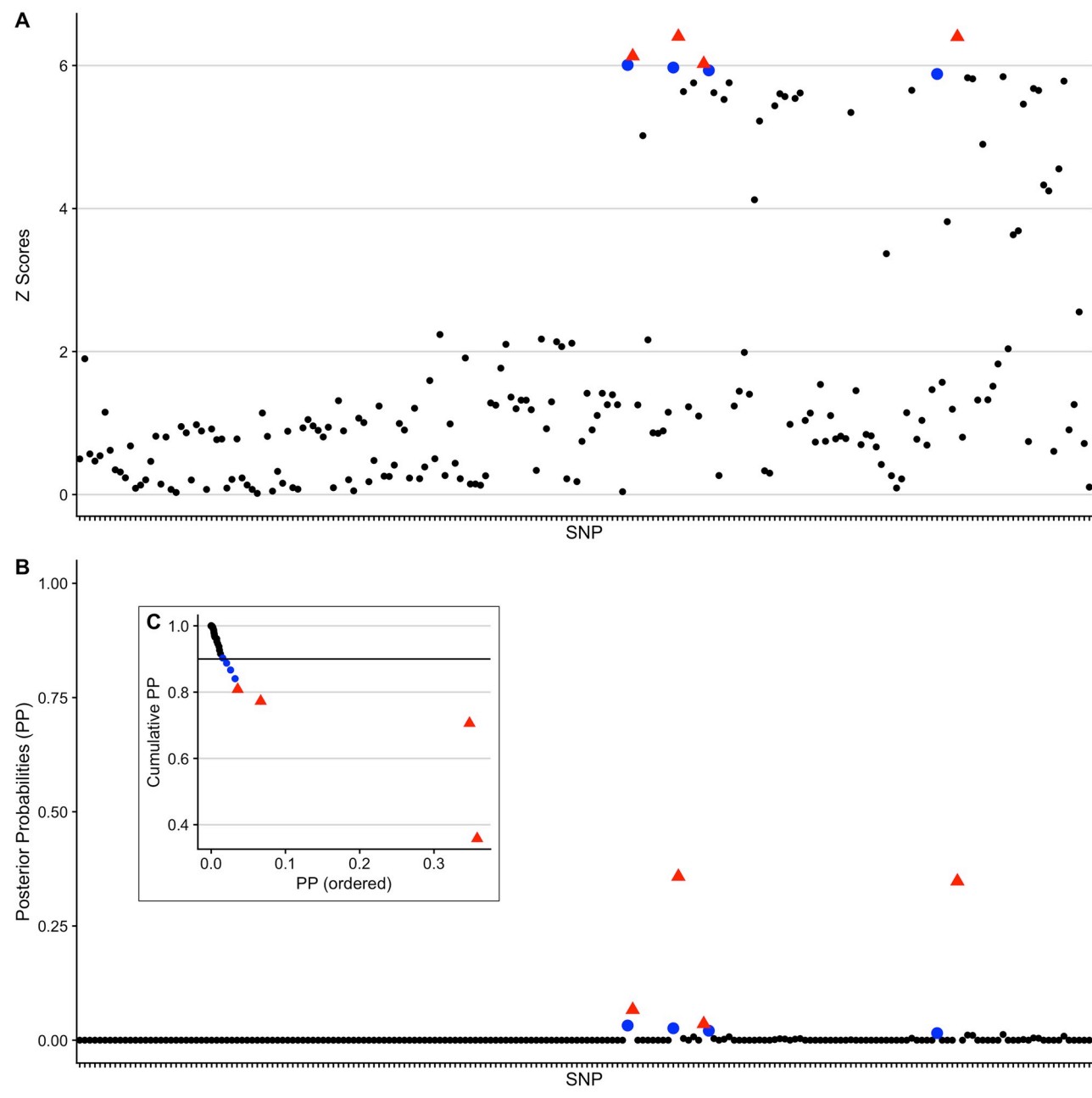

**Fig 4. A simple example to illustrate the results of the adjustment method.** (A) The absolute *Z* scores of the SNPs. (B) The PPs of the SNPs. (C) As in the fine-mapping procedure, variants are sorted into descending order of PP and summed. Starting with the SNP with the largest PP (far right) the cumulative sum (size) of the credible set is plotted as each SNP is added to the set. Red SNPs are those in the adjusted 90% credible set and blue SNPs are those that only appear in the original 90% credible set. The credible set formed of the red SNPs has an adjusted coverage estimate of 0.905 and the credible set formed of both the blue and red SNPs has an adjusted coverage estimate of 0.969.

adjusted because the threshold value (0.95) was within the 95% confidence interval for the adjusted coverage estimates of the original 95% credible sets, or because the credible set already contained only a single variant (and the adjusted coverage estimate was >0.95 implying that more variants did not need to be added to the set). Of the remaining 10 regions,

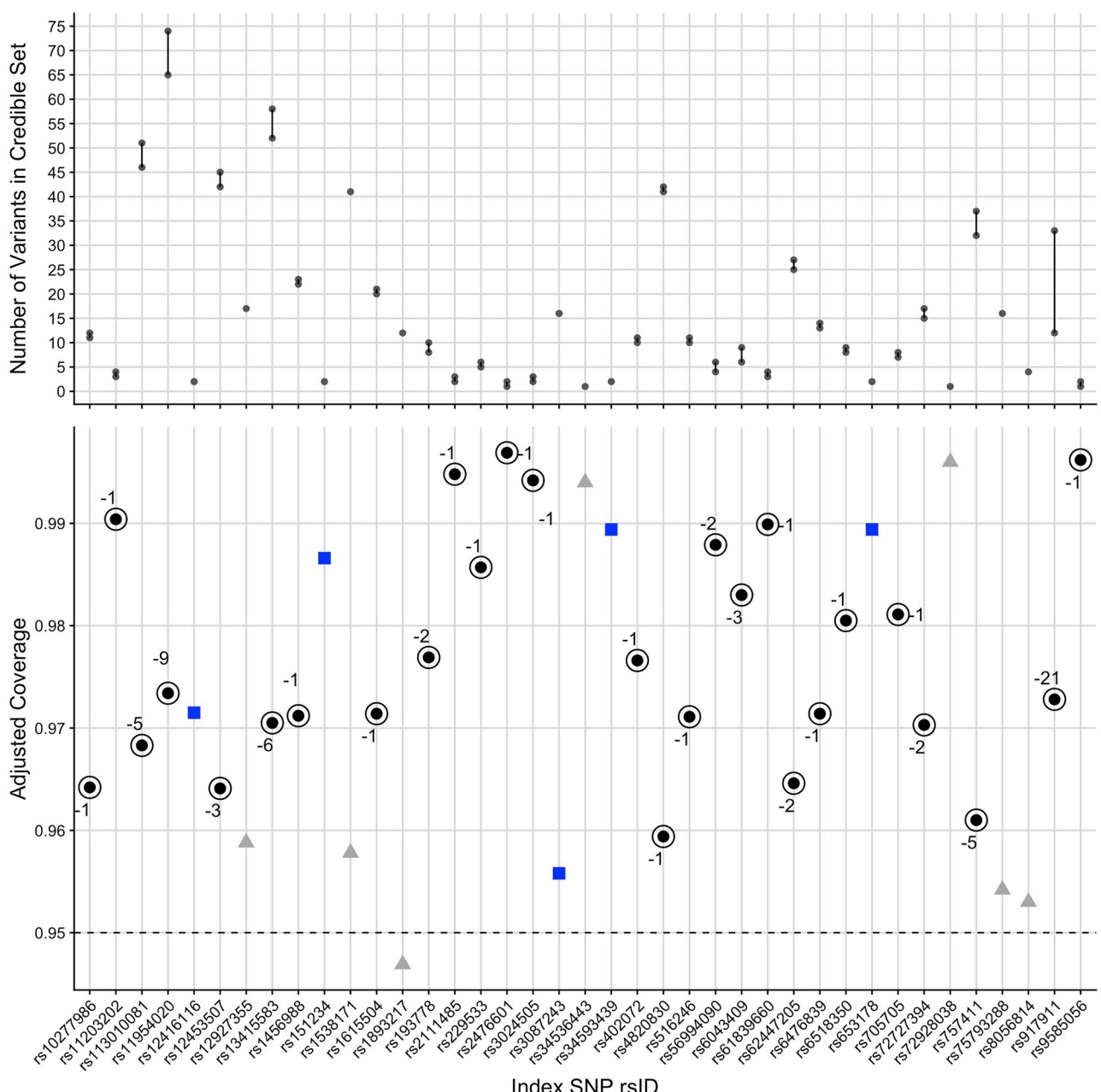

**Fig 5. Summary of adjusted coverage estimates and adjusted credible sets in T1D data set.** Top panel: The decrease in size of the credible set after adjustment. Bottom panel: The adjusted coverage estimates of 95% Bayesian credible sets for T1D-associated genomic regions. Black points represents regions where the credible set changed after the adjustment and the "-" values for the circled points represent the decrease in the number of variants from the standard to the adjusted 95% credible set. Blue points represent regions where the credible set did not change after the adjustment and grey points represent regions where the credible set did not need to be adjusted since the threshold (0.95) was contained in the 95% confidence interval of the conditional coverage estimate, or because the credible set already contained only a single variant.

4 credible sets did not change after adjustment and 6 credible sets did change after adjustment (dropping 1 variant). We observed that this happened in two ways (S12B Fig). In 3 cases, the adjusted coverage was greater than the claimed coverage, allowing a variant to be dropped. In the other 3 cases, the claimed coverage was near 1 (above 99% for a 95% target), and so there

was "space", with a tighter confidence interval about the adjusted coverage, for a lower target to be used whilst still maintaining adjusted coverage >95%. The full results for these 6 regions (with minimum $P < 10^{-12}$ and where the credible set changed after adjustment) are shown in S5 Table.

Fine-mapping to single base resolution has been used as a measure of GWAS resolution [22]. Two of the original 95% credible sets only contained a single variant: rs34536443 (missense in *TYK2*) and rs72928038 (intronic in *BACH2*). After applying our adjustment, two additional 95% credible sets were narrowed down from two variants to a single variant. First, rs2476601 (missense variant R620W in *PTPN22*) was selected, dropping rs6679677 which is in high LD with rs2476601 ($r^2 = 0.996$). These SNPs have high PPs (0.856185774 and 0.143814226, respectively) and the adjusted credible set containing only rs2476601 has an adjusted coverage estimate of 0.9501 with a 95% confidence interval of (0.9392, 0.9613).

Second, rs9585056 was selected, dropping rs9517719 ($r^2 = 0.483$). rs9517719 is intergenic, while rs9585056 is in the 3' UTR of the lncRNA *AL136961.1*, but has been shown to regulate expression of *GPR183* which in turn regulates an IRF7-driven inflammatory network [30]. While it is likely that R620W is indeed the causal variant at *PTPN22*, there is no conclusive data to evaluate whether rs9585056 is more likely to be causal compared to rs9517719. Nonetheless, the enrichment for missense variants is encouraging and in total, the number of putative causal variants for T1D in credible sets reduced from 658 to 582 upon adjustment.

## Discussion

Bayesian methods for single causal variant fine-mapping typically prioritise a credible set of putative causal variants. In this work, we have stratified by P value to show that the inferred probabilities that these credible sets do indeed contain the causal variant are systematically biased in simulations with intermediate power, reflecting those instances where fine-mapping is usually performed. We show that this is because fine-mapped regions are not a random sample as assumed by fine-mapping methods, but the subset of regions with the most extreme *P* values. Note that the effect of this censoring is actually most severe where *P* values just exceed the significance threshold, explaining why our adjustment has the most effect in the range of $10^{-12} < P < 10^{-4}$.

Threshold coverage estimates are typically conservatively biased, suggesting that the threshold of the credible set can be used as a lower bound for the true conditional coverage of the causal variant in the set [6, 21, 22]. In contrast, claimed coverage estimates (the sum of the PPs of the variants in the set) are anti-conservatively biased in low powered scenarios and conservatively biased in intermediately powered scenarios. We therefore developed an "adjusted coverage estimator" which has smaller expected error than the conventional conditional coverage estimates in the single causal variant literature, regardless of credible set threshold, power and the underlying LD structure in the region (and whether this is known or estimated from a reference panel).

The standard Bayesian approach for fine-mapping [9], and therefore our adjustment method, are limited in that they do not model multiple causal variants. Fine-mapping approaches that jointly model SNPs have been developed, such as GUESSFM [4] which uses genotype data and FINEMAP [10] and JAM [13] which attempt to reconstruct multivariate SNP associations from univariate GWAS summary statistics, differing both in the form they use for the likelihood and the method used to stochastically search the model space. The output from these methods are posterior probabilities for various configurations of causal variants, and therefore the grouping of SNPs to distinct association signals must often be performed post-hoc to obtain similar inferences to that of single causal variant fine-mapping.

Newer software versions (e.g. FINEMAP v1.3 and v1.4), however, are now able to produce lists of credible SNPs for the best multiple causal variant configurations. Thus, an interesting extension to this research would be to investigate whether the credible sets from multiple causal variant fine-mapping are subject to the conditional coverage biases exhibited in single-causal variant fine-mapping.

The sum of single effects (SuSiE) method [6] removes the single causal variant assumption and groups SNPs to distinct association signals in the analysis, such that it aims to find as many credible sets of variants that are required so that each set captures an effect variant, whilst also containing as few variants as possible (similar to "signal clusters" in Lee et al.'s DAP-G method [31]). This sophisticated approach has great potential, but the simulated 95% credible sets formed using both the SuSiE and DAP-G methods "typically had coverage slightly below 0.95, but usually >0.9" (Fig 3 and Fig S3 in [6]). Our method could potentially be extended to improve upon the conditional coverage of credible sets obtained using SuSiE and DAP-G fine-mapping methods.

Whilst our method does not address all the limitations of single causal variant fine-mapping, it improves upon the common inferences that are reported in the literature by researchers. We recommend that our adjustment is used as an extra step in the single causal variant fine-mapping pipeline, to obtain an adjusted coverage estimate that the causal variant is contained within the credible set and if required, to derive an adjusted credible set.

## Methods

### Design of simulation pipeline

We simulated a variety of genetic association studies using African and European haplotypes present in the 1000 Genomes Phase 3 data set [25]. Regions were selected that contained approximately 700 SNPs in low LD (African: Chr10:6030194-6219451), medium LD (European: Chr10:6030194-6219451) or high LD (European: Chr10:60969-431161) (GRCh37). In each simulation, causality was randomly allocated to one variant in the region and effect sizes were either (i) sampled from the prior ($\beta \sim N(0, 0.2^2)$) before being fixed at that sampled value to assess the empirical calibration of conditional coverage estimates, (ii) fixed at $\beta = log$ (1.05) or (iii) $\beta = log(1.2)$. Sample sizes (NN: number of cases = number of controls = 5000, 10000 or 50000) were also varied across simulations.

The haplotype frequencies and sampled parameter values were then used in the `simGWAS` R package [32] to: (i) simulate the results of a case-control GWAS (the study to "adjust") (ii) simulate results from 5000 case-control GWASs (to evaluate the accuracy of our method). These simulated GWAS results are marginal $Z$ scores, which were then converted to PPs using the `corrcoverage::ppfunc` or `corrcoverage::ppfunc.mat` functions, which are based on Maller et al.'s derivations (See 'Z scores to PPs' section below).

The variants are sorted into descending order of their PPs and these are summed until the credible set threshold (0.9 or 0.95) is exceeded. The variants required to exceed this threshold comprise the 90% or 95% credible set. The sum of the PPs of the variants in the credible set is the "claimed coverage" [23], which must be greater than or equal to the threshold by virtue of the method. The frequentist empirical estimate of the true conditional coverage is calculated as the proportion of 5000 simulated credible sets that contain the causal variant (CV). The adjusted coverage estimate is also calculated for each credible set using the `corrcoverage::corrcov` function. The simulation procedure is repeated many times to obtain a final simulation data set, consisting of the sampled parameter values and the empirical, claimed and adjusted conditional coverage estimates of the simulated credible sets.

For the evaluation of averaging results over a range of LD patterns, we used haplotypes from the UK10K data. In each simulation, genomic regions were randomly selected (bounded by recombination hotspots defined using the LD detect method [33]) on chromosome 22 and two non-overlapping sets of 100 adjacent variants were selected, so that the simulated region consisted of 200 correlated and non-correlated variants. The simulation pipeline described above was then followed to obtain a final simulation data set for various LD patterns.

For investigating the effect of violating the single causal variant assumption, 2 CVs were simulated in each genomic region, which were either in high LD ($r^2 > 0.7$) or low LD ($r^2 < 0.01$), and conditional coverage was defined as the frequency with which a credible set contained at least one of the CVs. The odds ratio of the simulated CVs were sampled independently and sample sizes were varied so that the power of the simulated systems varied (S13 Fig). If multiple CVs are present in a region then multiple possible definitions of conditional coverage exist, and as the number of CVs in a region increases, the number of possible definitions and the discrepancies in their interpretation increases. For this reason, we chose not to analyse conditional coverage properties in regions containing >2 CVs.

## *Z* scores to PPs

Maller et al. derive a method to calculate PPs from GWAS summary statistics (Supplementary text in [9]) upon which the following is based. Let $\hat{\beta}_i$ for $i = 1, \ldots, k$ SNPs in a genomic region be the regression coefficient from a single-SNP logistic regression model, quantifying the evidence of an association between SNP $i$ and the disease. Assuming that there is only one CV per region and that this is typed in the study, if SNP $i$ is causal, then the underlying log odds ratio $\beta_i \neq 0$ and $\beta_j$ (for $j \neq i$) is non-zero only through LD between SNPs $i$ and $j$. Note that no parametric assumptions are required for $\beta_i$ yet, so we write simply that it is sampled from some distribution, $\beta_i \sim [\ ]$. The likelihood is then,

$$
\begin{aligned}
P(D|\beta_i \sim [\ ], \ i \ \text{causal}) \quad &= P(D_i|\beta_i \sim [\ ], \ i \ \text{causal}) \times P(D_{-i}|D_i, \ \beta_i \sim [\ ], \ i \ \text{causal}) \\
&= P(D_i|\beta_i \sim [\ ], \ i \ \text{causal}) \times P(D_{-i}|D_i, \ i \ \text{causal}) ,
\end{aligned}
\tag{1}
$$

since $D_{-i}$ is independent of $\beta_i$ given $D_i$. Here, $D$ is the genotype data (0, 1 or 2 counts of the minor allele per individual) for all SNPs in the genomic region and $i$ is a SNP in the region, such that $D_i$ and $D_{-i}$ are the genotype data at SNP $i$ and at the remaining SNPs, respectively.

Parametric assumptions can now be placed on SNP $i$'s true effect on disease. This is typically quantified as log odds ratio, and is assumed to be sampled from a Gaussian distribution, $\beta_i \sim N(0, W)$, where $W$ is chosen to reflect the researcher's prior belief on the variability of the true OR. Conventionally $W$ is set to $0.2^2$, reflecting a belief that there is 95% probability that the odds ratio at the causal variant lies between $exp(-1.96 \times 0.2) = 0.68$ and $exp(1.96 \times 0.2) = 1.48$.

The posterior probabilities of causality for each SNP $i$ in an associated genomic region with $k$ SNPs can be calculated as,

$$
PP_i = P(\beta_i \sim N(0, W), \ i \ \text{causal}|D) , \quad i \in \{1, ..., k\}.
\tag{2}
$$

Under the assumption that each SNP is *a priori* equally likely to be causal, then

$$
P(\beta_i \sim N(0, W), \ i \ \text{causal}) = \frac{1}{k} , \quad i \in \{1, ..., k\}
\tag{3}
$$

and Bayes theorem can be used to write

$$PP_i = P(\beta_i \sim N(0, W), \ i \ \text{causal}|D) \propto P(D|\beta_i \sim N(0, W), \ i \ \text{causal}). \tag{4}$$

Dividing through by the probability of the genotype data given the null model of no genetic effect, $H_0$, yields a likelihood ratio,

$$PP_i \propto \frac{P(D|\beta_i \sim N(0, W), \ i \ \text{causal})}{P(D|H_0)}, \tag{5}$$

from which Eq (1) can be used to derive,

$$PP_i \propto \frac{P(D_i|\beta_i \sim N(0, W), \ i \ \text{causal})}{P(D_i|H_0)} = BF_i, \tag{6}$$

where $BF_i$ is the Bayes factor for SNP $i$, measuring the ratio of the probabilities of the data at SNP $i$ given the alternative (SNP $i$ is causal) and the null (no genetic effect) models.

In genetic association studies where sample sizes are usually large, these BFs can be approximated using Wakefield's Approximate Bayes Factor (ABF) derivation [34]. Given that $\hat{\beta}_i \sim N(\beta_i, V_i)$ and *a priori* $\beta_i \sim N(0, W)$,

$$PP_i \propto BF_i \approx ABF_i = \sqrt{\frac{V_i}{V_i + W}} exp\left(\frac{Z_i^2}{2}\frac{W}{(V_i + W)}\right), \tag{7}$$

where $Z_i^2 = \frac{\hat{\beta}_i^2}{V_i}$ is the squared marginal $Z$ score for SNP $i$.

If $V_i$ is unknown, we approximate it using

$$V_i = \frac{1}{2 \times N \times MAF_i \times (1 - MAF_i) \times s \times (1 - s)} \tag{8}$$

where $N$ is the total sample size, $s$ is the proportion of cases and $MAF_i$ is the $MAF$ for SNP $i$.

## Distribution of marginal $Z$ scores under a single CV model

Associations between a SNP and a trait are usually tested for using single-SNP models, such that marginal $Z$ scores are derived. In contrast, if the SNPs in the region are jointly modelled, then joint $Z$ scores can be derived. Under the assumption of a single CV per region, we can write down the expected joint $Z$ score vector,

$$Z_J = (0, \cdots, 0, \mu, 0, \cdots, 0)^T, \tag{9}$$

where $Z_J$ has length equal to the number of SNPs in the region, and all elements take the value 0 except at the position of the causal variant, which takes the value $\mu$.

Given $Z_J$, the expected marginal $Z$ scores can be written as

$$E(Z) = \Sigma \times Z_J, \tag{10}$$

where $\Sigma$ is the SNP correlation matrix [35]. The asymptotic distribution of these marginal $Z$ scores is then multi-variate normal (MVN) with variance equal to the SNP correlation matrix [35],

$$Z \sim MVN(E(Z), \Sigma). \tag{11}$$

## Adjusted coverage estimate

The value of $\mu$ is unknown in genetic association studies and it is therefore estimated in our method to derive the $Z_J$ vector. We consider using the absolute $Z$ score at the lead-SNP as an estimate for $\mu$, but find this to be too high in low powered scenarios (S6 Fig). This is because $E(|Z|) > 0$ even when $E(Z) = 0$, and thus $E(|Z|) > E(Z)$ when $E(Z)$ is close to 0. Instead, we consider a weighted average of the absolute $Z$ scores, so that for a region comprising of $k$ SNPs,

$$\hat{\mu} = \sum_{i=1}^{k} |Z_i| \times PP_i. \tag{12}$$

We find this estimate to have small relative error even at small $\mu$ (S6 Fig).

Given $\hat{\mu}$, we consider each SNP $i$ in the region as the CV in turn, and construct the joint $Z$ vector as

$$\hat{Z}_J[j] = \begin{cases} 0 & j \neq i \\ \hat{\mu} & j = i \end{cases} \tag{13}$$

We simulate $N = 1000$ marginal $Z$ score vectors,

$$\mathcal{Z}_{N=1000}^* = \{Z_1^*, \ldots, Z_{1000}^*\} \overset{iid}{\sim} MVN(\Sigma \times \hat{Z}_J, \Sigma). \tag{14}$$

Each simulated $Z^*$ vector is then converted to PPs and credible sets are formed using the standard Bayesian method (sort and sum). The proportion of the $N = 1000$ simulated credible sets that contain SNP $i$, $prop_i$, is calculated.

This procedure is implemented for each SNP in the genomic region with $PP > 0.001$ (this value can be altered using the 'pp0min' parameter in the software) considered as causal. The final adjusted coverage estimate is then calculated by weighting each of these proportions by the PP of the SNP considered causal,

$$\text{Adjusted Coverage Estimate} = \frac{\sum_{i:PP_i>0.001} PP_i \times prop_i}{\sum_{i:PP_i>0.001} PP_i}. \tag{15}$$

Note that we are not attempting to reweight the PPs for inference, only to calibrate the adjusted coverage estimate. Intuitively, proportions obtained from realistic scenarios (SNPs with high posterior probabilities of causality considered as causal) are up-weighted and proportions obtained from more unrealistic scenarios (SNPs with low posterior probabilities of causality considered as causal) are down-weighted.

A value of $N = 1000$ (so that 1000 credible sets are simulated for each SNP that is considered causal) was found to be a robust choice, but is included as an optional parameter in the software. This allows users to increase or decrease the value as desired, for example in the interest of computational time for small or large numbers of SNPs in a genomic region, respectively.

## Using a reference panel for MAF and LD

We evaluated the performance of adjusted coverage estimates when using a reference population to approximate MAFs and SNP correlations. In this analysis, we selected an LD block (chr10:6030194-6219451) and chose only the SNPs in this region that could be matched by their position between the 1000 Genomes data and the UK10K data (578 SNPs) for our simulations. European haplotype data for these SNPs were collected from both the 1000 Genomes data (consisting of 503 individuals) and the UK10K data (consisting of 3781 individuals).

As in our standard simulation pipeline, causality was randomly allocated to one of these variants with its effect size either sampled from $N(0, 0.2^2)$ or fixed. Sample sizes (NN: number of cases = number of controls = 5000, 10000 or 50000) were also varied across simulations. These sampled parameter values were then used with MAFs and haplotype data from the 1000 Genomes data to simulate marginal $Z$ scores from various genetic association studies. The standard claimed and adjusted coverage estimates (Fig 2A and 2B respectively) were then derived as usual and the adjusted coverage estimates were also calculated when using the reference data to estimate MAFs and SNP correlations (Fig 2C).

For comparison, we also investigated the effect of using a reference panel for the adjustment in the high LD region previously discussed (we omitted the low LD region as this used African haplotypes, for which we do not have a large representative reference panel). The results were similar, indicating that there is minimal loss of accuracy in adjusted coverage estimates when approximating SNP correlations using a reference panel (S8 Fig).

### T1D data set

For the T1D data analysis, we used the index SNPs for the genomic regions reported in the original study [29] and used Immunochip data to find the other SNPs in each of these regions. We then used the `corrcoverage` R package with default parameters to find 95% (and 99%) credible sets of variants, along with the claimed and adjusted coverage estimates for each of them. 95% confidence intervals for the adjusted coverage estimates were derived by calculating 100 adjusted coverage estimates and taking the 2.5th and 97.5th percentile of them. If 0.95 (or 0.99 for 99% credible sets) did not fall within this confidence interval, then the `corrcoverage::corrected_cs` function (with the following optional parameter values: 'acc = 0.0001, max.iter = 70') was used to find an adjusted credible set; that is, the smallest set of variants required such that the adjusted coverage of the resultant credible set is close to the threshold value (within 0.0001 or as close as possible within 70 iterations).

## Supporting information

**S1 Fig. Distributions of absolute effect sizes.** (A) Histogram of absolute effect sizes sampled from the prior, $\beta \sim N(0, 0.2^2)$ (B) Histogram of absolute effect sizes of the lead-SNP in genome-wide significantly associated regions from case-control studies deposited on the GWAS catalog. Blue curve overlaid is for $N(0, 0.2^2)$ distribution, black dashed line is where $\beta = log(1.05)$ and red dotted line is where $\beta = log(1.2)$. The $x$ axis has been truncated to remove extreme values. The distributions are quite different, resulting from censoring of smaller $\beta$ which are less likely to be associated with a genome-wide significant P value (see Fig 1).
(TIF)

**S2 Fig. Error of conditional coverage estimates for 90% credible sets.** Error is calculated as estimated conditional coverage–empirical conditional coverage where empirical conditional coverage is the proportion of 5000 replicate credible sets that contain the causal variant. Box plots showing error in conditional coverage estimates for 5000 (A) low (B) medium and (C) high LD simulations. Conditional coverage estimates are the threshold (0.9) (left), the claimed coverage (the sum of the posterior probabilities of the variants in the credible set) averaged over all simulations (left-middle) or for simulations binned by minimum P value in the region (right-middle) and the adjusted coverage estimate (right) binned by minimum P value in the region. Black diamond shows mean error. Two simulations for $\beta = log(1.05)$

simulations that fell into $(10^{-12}, 0]$ bin were manually removed as a box plot could not be generated. Graphical display of SNP correlation matrix for each region shown.
(TIF)

**S3 Fig. Error of conditional coverage estimates for 95% credible sets.** Error is calculated as estimated conditional coverage–empirical conditional coverage where empirical conditional coverage is the proportion of 5000 replicate credible sets that contain the causal variant. Box plots showing error in conditional coverage estimates for 5000 (A) low (B) medium and (C) high LD simulations. Conditional coverage estimates are the threshold (0.95) (left), the claimed coverage (the sum of the posterior probabilities of the variants in the credible set) averaged over all simulations (left-middle) or for simulations binned by minimum P value in the region (right-middle) and the adjusted coverage estimate (right) binned by minimum P value in the region. Black diamond shows mean error. Two simulations for $\beta = log(1.05)$ simulations that fell into $(10^{-12}, 0]$ bin were manually removed as a box plot could not be generated. Graphical display of SNP correlation matrix for each region shown.
(TIF)

**S4 Fig. Error of conditional coverage estimates for 90% credible sets split up by sample size.** Error is calculated as estimated conditional coverage–empirical conditional coverage where empirical conditional coverage is the proportion of 5000 replicate credible sets that contain the causal variant. Box plots showing error in conditional coverage estimates for 5000 simulations with $N0$ (number of controls) = $N1$ (number of cases) = (A) 5000 (B) 10000 and (C) 50000. Overall error in (left) threshold and (middle) claimed coverage estimates averaged across all 5000 simulations. Right hand plots show error in claimed coverage estimates for different P value bins. If there were <10 simulations contained in a P value bin, then these were manually removed (for example in $P_{min} < 10^{-6}$ bins for $\beta = log(1.05)$, $N0 = N1 = 5000$).
(TIF)

**S5 Fig. Error of conditional coverage estimates for credible sets using UK10K data.** Box plots showing the error, estimated conditional coverage–empirical conditional coverage, of (A) 90% and (B) 95% credible sets where empirical conditional coverage is the proportion of 5000 replicate credible sets that contain the causal variant. Error of conditional coverage estimates where (left) conditional coverage estimate equals threshold, (middle) conditional coverage estimate equals claimed coverage (sum of the posterior probabilities of the variants in the set) and (right) conditional coverage estimate is adjusted coverage. Results from 5000 simulations for each simulation type have been averaged over many genomic regions that vary in LD patterns.
(TIF)

**S6 Fig. Estimating $\mu$.** Error of $\mu$ estimates calculated as $\hat{\mu}_X - \mu$. The $x$ axis is the joint $Z$ score at the CV. Line is fitted using a GAM as the smoothing function (geom_smooth() in `ggplot2`). (A) $\hat{\mu} = \max_{i \in \{1,\dots,k\}} (|Z_i|)$ (B) $\hat{\mu} = \sum_{i=1}^{k} |Z_i| \times PP_i$.
(TIF)

**S7 Fig. Error of conditional coverage estimates for 90% credible sets for various values of $W$.** Error is calculated as estimated conditional coverage–empirical conditional coverage where empirical conditional coverage is the proportion of 5000 replicate credible sets that contain the causal variant. Box plots showing error in conditional coverage estimates for 5000 medium LD simulations. Conditional coverage estimates are the claimed coverage (top) and the adjusted coverage estimate (bottom) for simulations binned by minimum P value in the region. Black diamond shows mean error.
(TIF)

**S8 Fig. Error of conditional coverage estimates for 90% credible sets using a reference panel to approximate MAFs and SNP correlations in a high LD region.** Error is calculated as estimated conditional coverage–empirical conditional coverage. Conditional coverage estimates from 5000 simulations using original 1000 Genomes data and UK10K data as a reference panel. (A) Claimed coverage estimate (the sum of the posterior probabilities of causality for the variants in the credible set) (B) Adjusted coverage estimate (C) Adjusted coverage estimate using UK10K data to approximate MAFs and SNP correlations (D) Graphical display of SNP correlations in 1000 Genomes data (E) Graphical display of the estimated SNP correlations in UK10K data. Two simulations for $\beta = log(1.05)$ simulations that fell into $(10^{-12}, 0]$ bin were manually removed as a box plot could not be generated.
(TIF)

**S9 Fig. Empirical estimate of the true conditional coverage of adjusted 90% and 95% credible sets.** 100,000 simulated (A) 90% and (B) 95% credible sets were adjusted using the `corr-coverage::corrected_cs` function (with default parameters and 'desired.cov = 0.9' or 'desired.cov = 0.95'), and the "required threshold" value obtained from each simulation was used to form 5000 replicate credible sets to estimate the empirical conditional coverage of these adjusted (A) 90% and (B) 95% credible sets.
(TIF)

**S10 Fig. R package timings.** Curve showing the timings of the `corrcoverage::corrcov` function for different sized genomic regions. For each size of genomic region analysed, 50 replicates of the `corrcoverage::corrcov` function were ran and the mean time taken is plotted. Curve drawn using geom_smooth() function in `ggplot2`. Simulations ran using one core of an Intel Xeon Gold 6142 processor running at 2.6GHz.
(TIF)

**S11 Fig. Summary of adjusted coverage estimates and adjusted 99% credible sets in T1D data set.** Top panel: The decrease in size of the credible set after adjustment. Bottom panel: The adjusted coverage estimates of 99% Bayesian credible sets for T1D-associated genomic regions. Black points represents regions where the credible set changed after the adjustment and the "-" values for the circled points represent the decrease in the number of variants from the standard to the adjusted 99% credible set. Blue points represent regions where the credible set did not change after the adjustment and grey points represent regions where the credible set did not need to be adjusted since the threshold was contained in the 99% confidence interval of the coverage estimate, or because the credible set already contained only a single variant.
(TIF)

**S12 Fig. Further analysis of T1D results.** (A) Box plots showing the distribution of lead-SNP $-log10(P)$ values for genomic regions where the 95% credible set changed after adjustment or where it did not change after adjustment. (B) Adjusted coverage estimates against claimed coverage estimates of standard 95% credible sets for 39 genomic regions associated with T1D. Red points are those with lead-SNP $P < 10^{-12}$ and where the credible set changed after adjustment, blue points are those with lead-SNP $P < 10^{-12}$ and where the credible set did not change after adjustment and green points are those with lead-SNP $P < 10^{-12}$ whose 95% credible set did not need to be adjusted.
(TIF)

**S13 Fig. Distribution of the minimum *P* value for 2 CV simulations (Fig 2).** 2 CVs are (A) in low LD ($r^2 < 0.01$) (B) in high LD ($r^2 > 0.7$). Faceted by odds ratio values at the causal variants. (TIF)

**S1 File. Individual plots for 95% credible set T1D analysis.** Zip file containing Z-score plots, PP plots and Manhattan plots for the 39 T1D association regions analysed. (ZIP)

**S2 File. Individual plots for 99% credible set T1D analysis.** Zip file containing Z-score plots, PP plots and Manhattan plots for the 39 T1D association regions analysed. (ZIP)

**S1 Table. T1D adjusted 95% credible set results.** (CSV)

**S2 Table. List of 95% credible sets before and after adjustment.** (CSV)

**S3 Table. T1D adjusted 99% credible set results.** (CSV)

**S4 Table. List of 99% credible sets before and after adjustment.** (CSV)

**S5 Table. Full results for the 6 genomic regions with $P < 10^{-12}$ and where the 95% credible set changed after adjustment.** (CSV)

## Acknowledgments

We thank Paul Newcombe and Rob Goudie for helpful discussions, and Kevin Kunzmann for advice on creating R packages.

## Author Contributions

**Conceptualization:** Anna Hutchinson, Chris Wallace.

**Data curation:** Anna Hutchinson.

**Formal analysis:** Anna Hutchinson.

**Investigation:** Anna Hutchinson, Hope Watson, Chris Wallace.

**Methodology:** Anna Hutchinson, Chris Wallace.

**Resources:** Anna Hutchinson.

**Software:** Anna Hutchinson, Chris Wallace.

**Supervision:** Chris Wallace.

**Validation:** Anna Hutchinson.

**Visualization:** Anna Hutchinson.

**Writing – original draft:** Anna Hutchinson, Chris Wallace.

**Writing – review & editing:** Hope Watson, Chris Wallace.

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
