## [Decision Letter · Decision Letter 0]

18 Feb 2020

Dear Miss Hutchinson,

Thank you very much for submitting your manuscript "Improving the coverage of credible sets in Bayesian genetic fine-mapping" for consideration at PLOS Computational Biology. As with all papers reviewed by the journal, your manuscript was reviewed by members of the editorial board and by several independent reviewers. The reviewers appreciated the attention to an important topic. Based on the reviews, we are likely to accept this manuscript for publication, providing that you modify the manuscript according to the review recommendations.

Sincerely,

Ilya Ioshikhes

Associate Editor

PLOS Computational Biology

Jian Ma

Deputy Editor

PLOS Computational Biology

[LINK]

Reviewer's Responses to Questions

**Comments to the Authors:**

Reviewer #1: The authors argue that Bayesian credible sets of causal SNPs, computed during fine mapping of GWAS hits, tend to overestimate their coverage. Bias arises both by the fixed nature of a genetic effect at a given locus, and from selection of regions for fine mapping according to statistically significant association. An algorithmic method is proposed to estimate the magnitude of bias and adjust the coverage probability accordingly. An algorithm is presented to re-estimate credible sets with pre-specified coverage, and is shown to reduce the size of credible sets in several regions associated with type-1 diabetes. The paper raises some interesting points and should be of broad interest. It is refreshingly well written and low on typos.

Major point

1. The paper hinges on the understanding of the term “coverage”. While a frequentist understanding is alluded to in the introduction, the authors take a particular interpretation that may not be universally held. Specifically, the authors ask how often the credible set for the specific region under study covers the causal variant, given that it has been followed up. Here the fixed effects view is justifiable, and the bias of the standard calculation is shown (fig 1A right). However, given that credible sets are generated for many regions, both within and between studies (as the authors have done so themselves in the applied example), we can equally ask how often the credible set covers the causal variant, on average across all regions. This is an equally valid frequentist version of the question “what is the probability that the causal variant is contained in the credible set?” Here there is much less bias (fig 1A left). The bias by selecting significant regions is also open to interpretation. For regions not followed up, we effectively have estimated PP of zero for all SNPs and an empty credible set, which does not include the causal variant. Averaging over these regions as well as those explicitly followed up, the coverage probabilities would be unbiased. Although not a declared Bayesian, I am inclined to favour these broader understandings of coverage since the random effects scenarios are more common than the repeated sampling ones in practice. The authors are welcome to their view but should discuss this key point of interpretation in the introduction and discussion. I also suggest a change in terminology from “corrected coverage estimate” to “conditional coverage estimate” to reflect that it is only a correction under an interpretation that conditions on the specific region, phenotype and population being studied.

Minor points

2. P5 L139 “effect size” – it’s an expected Z-score that is derived, which is not the same as an effect size.

3. P5 L151 “Our method relies on MAF and SNP correlation data” – I don’t see how MAF information is needed to simulate correlated Z scores. It was used to simulate test data for the method, but doesn’t seem necessary for the method itself.

4. P7 L222 “the claimed coverage estimates are too low” – this seems to be based simply on comparing the sizes of the standard and corrected credible sets. Without knowledge of the actual causal variants, we can’t be sure that the corrected sets have the desired coverage.

5. In fig S1 the N(0,0.2^2) actually looks like quite a poor fit to the GWAS data. Would smaller bin sizes improve the visual fit, or is this just a poor model? The choice of prior for the ABF is another factor that could bias the coverage estimate, and this ought to be discussed.

Typos

6. P2 L10 “loci” -> “locus”

7. P12 L406 “was” -> “were”

8. P12 L410 “it’s” -> “its”

9. P13 L429 and L431 “these” -> “them”

Reviewer #2: This was an interesting and topical manuscript. My overall enthusiasm for the manuscript was slightly diminished by the fact it was unclear how serious a problem the issue raised (namely that of the conservative coverage of fine-mapped variants) would be in practice. In particular, Figure 1C (rightmost plot) suggested that for signals reaching the de-facto genome-wise significance threshold of around 10-8, there is in fact little discrepancy between the claimed and true coverage.

Given this fact, I am slightly unclear which analyses presented in the manuscript support the statements made in the Abstract: “We show that this is because fine-mapping data sets are not randomly selected from amongst all causal variants, but from amongst causal variants with larger effect sizes” and in the Discussion: “We show that this is because fine mapped regions are not a random sample as assumed by fine mapping methods, but the subset of regions with the most extreme P values.”? Surely Figure 1C (rightmost plot) suggests that the regions with the most extreme P values (which will likely correspond to those with the larger effect sizes) are exactly those regions that will NOT be affected by the problem?! As these regions in fact show little discrepancy between the claimed and true coverage?

Additional comments:

Page 2, Line 23: reference [7] – it might also be worth referencing Ayers and Cordell (2010) https://onlinelibrary.wiley.com/doi/full/10.1002/gepi.20543 here, who I think also investigated the use of penalised regression for fine mapping (prior to Valdar et al. 2012)?

Page 3, lines 70-81 (see also page 9 lines 293-294): this issue of simulating from the prior as opposed from a fixed beta requires more detailed discussion/explanation. By “simulating from the prior” do you mean you resample the value of beta within each simulation replicate? I cannot see this as a correct approach (though I don’t mind you presenting it, provided it is made clear that it is not correct!) Within any given data set, the effect size at any particular causal variant surely takes a specific value (that may be considered to have been drawn from some distribution before the study started). If one is trying to investigate the properties of some analysis approach under repeated sampling of the dataset, surely one should indeed leave the effect size at the variant fixed at that sampled value?

This is a bit different from the idea of simulating the effect sizes of multiple different causal variants across the genome from a distribution (a model that underpins a lot of the recent work on SNP=based heritability estimation, linear mixed models, GCTA etc.), Depending on context one might want to resample the random effects, although I think would argue that for analysis of a specific data set, the effects (one sampled) should be considered fixed. It is also a bit different from a random effects meta-analysis whereby one could consider different studies to have genuinely different effect sizes (albeit sampled from a distribution with a common mean). Anyway, I think this all needs a bit more clarification and discussion.

Page 6, lines 190-191: This finding (that a corrected 90% credible set could be constructed using a threshold value of 0.781) is presumably specific to this particular example? i.e. this calculation would need to be repeated for any given GWAS that one was analysing, if one wished to calculate the appropriate threshold? Could be clarified.

Page 7 lines 215-225: It would be informative to state what significance levels were achieved in the 40 genomic regions interrogated. Given that your approach is successful in reducing the size of the 95% and 99% credible sets, it suggests there must indeed be some degree of conservative coverage for this data set when using standard approaches. This, combined with the observation made above (that for signals reaching the de-facto genome-wise significance threshold of around 10-8, there is in fact little discrepancy between the claimed and true coverage) suggests that many of the the 40 regions must not in fact have reached genome-wide significance levels?

Page 8 lines 261-270: You are correct that methods such as GUESSFM and FINEMAP and JAM produce more complicated outputs in terms of posterior probabilities for various configurations of causal variants. However I am not sure that *that much* post-processing is necessary to investigate these approaches. In particular, FINEMAP versions 1.3 and 1.4 already produces lists of credible SNPs for the best single-SNP configuration, the best two-SNP configuration, the best 3-SNP configuration etc. etc. It would certainly be interesting to investigate whether these lists of credible SNPs are subject to the same issue that you have highlighted for single-variant fine mapping approaches.

Page 10 line 326-327: I am not sure that I follow why defining coverage for regions containing > 2 CVs is particularly problematic? Would you not just define it as the frequency with which a credible set contained at least one of the CVs (as you did for regions with 2 CVs)? I could imagine that the more CVs there are in a region, the easier it would be for *any one* of them to “get chosen” into the credible set – I’m not sure what effect this would have on the coverage? However this issue would presumably pertain to the 2 CV situation as well?

Page 11 line 368: Rephrase “at the causal SNP’s position which takes the value $\\mu$” as “at the causal SNP’s position where the value of $z_i$ takes the value $\\mu$” ? (That is provided you define the $i$th element of $Z_J$ as $z_i$, which you might need to do earlier).

**Have all data underlying the figures and results presented in the manuscript been provided?**

Reviewer #1: Yes

Reviewer #2: Yes

PLOS authors have the option to publish the peer review history of their article (what does this mean?). If published, this will include your full peer review and any attached files.

Reviewer #1: No

Reviewer #2: No
---

## [Decision Letter · Decision Letter 1]

27 Mar 2020

Dear Miss Hutchinson,

We are pleased to inform you that your manuscript 'Improving the coverage of credible sets in Bayesian genetic fine-mapping' has been provisionally accepted for publication in PLOS Computational Biology.

Best regards,

Ilya Ioshikhes

Associate Editor

PLOS Computational Biology

Jian Ma

Deputy Editor

PLOS Computational Biology

Reviewer's Responses to Questions

**Comments to the Authors:**

Reviewer #1: I'm satisfied with these revisions. Thanks.

Reviewer #2: The authors have satisfactorily addressed all my previous comments.

**Have all data underlying the figures and results presented in the manuscript been provided?**

Reviewer #1: None

Reviewer #2: Yes

PLOS authors have the option to publish the peer review history of their article (what does this mean?). If published, this will include your full peer review and any attached files.

Reviewer #1: No

Reviewer #2: No

---

## [Editor Report · Acceptance letter]

3 Apr 2020

PCOMPBIOL-D-20-00068R1 

Improving the coverage of credible sets in Bayesian genetic fine-mapping

Dear Dr Hutchinson,

I am pleased to inform you that your manuscript has been formally accepted for publication in PLOS Computational Biology. Your manuscript is now with our production department and you will be notified of the publication date in due course.

With kind regards,

Laura Mallard
